# High PEEP Increases Airway Dead Space and Decreases Alveolar Ventilation: A New Technique for Volumetric Capnography

**DOI:** 10.3390/biomedicines13092275

**Published:** 2025-09-16

**Authors:** Masashi Zuiki, Kazunori Watanabe, Norihiro Iwata, Rika Mitsuno, Madoka Konishi, Akio Yamano, Eisuke Ichise, Hidechika Morimoto, Kanae Hashiguchi, Tatsuji Hasegawa, Tomoko Iehara

**Affiliations:** Department of Paediatrics, Kyoto Prefectural University of Medicine, Kyoto 602-0841, Japan

**Keywords:** capnography, positive end-expiratory pressure, newborn, dead space volume

## Abstract

**Background/Objectives:** Identifying the optimal positive end-expiratory pressure (PEEP) is a major challenge in implementing strategies to prevent ventilator-induced lung injury in newborns. In this study, we assessed the validity of volumetric capnography based on the neonatal patient monitor (V_cap,PM_) technique and investigated the impact of PEEP on newborns. **Methods:** Analysis 1 evaluated the validity of the V_cap,PM_ technique with data from pediatric patients receiving invasive respiratory support. Linear regression and Bland–Altman analyses were performed on V_cap,PM_ and HAMILTON-C1 data. Analysis 2 evaluated the impact of PEEP on newborns. The PEEP level was increased from mild to high (the incremental phase) and then decreased from high to mild (the decremental phase) while performing the V_cap,PM_ technique on term and preterm infants. **Results:** Analysis 1 included 31 children (age, 9 [interquartile range (IQR), 0–36] months; weight, 6.0 [IQR, 3.8–10.5] kg). Regression and Bland–Altman analyses demonstrated the accuracy of V_cap,PM_. Analysis 2 included 28 term (mean gestational age, 38 [IQR, 38–40] weeks; weight, 2924 [IQR, 2725–3109] g) and 21 preterm (mean gestational age, 33 [IQR, 31–34] weeks; weight, 1918 [IQR, 1356–2186] g) newborns. Despite no difference in tidal volume, high PEEP significantly increased airway dead space and decreased alveolar tidal volume compared to mild PEEP in each phase in term and preterm neonates. **Conclusions:** High PEEP induced airway dilation in newborns, as determined using a novel V_cap_ technique. This technique, which requires no special equipment, has the potential for wider clinical application in neonatal care.

## 1. Introduction

Newborns admitted to the neonatal intensive care unit (NICU) often require respiratory support. Considering the potential for ventilator-induced lung injury (VILI) in underdeveloped lungs, it is critical to implement lung-protective strategies, including selecting the appropriate level of positive end-expiratory pressure (PEEP) for newborns [1,2]. Currently, the biggest challenge in implementing this strategy is identifying the optimal PEEP to prevent end-expiratory lung collapse and alveolar overinflation, the two primary factors contributing to VILI [3,4]. Additionally, an incorrect PEEP may result in hemodynamic instability due to right ventricular dysfunction [5,6]. However, there is insufficient evidence to guide the selection of the appropriate PEEP level in premature infants receiving mechanical ventilation due to the inadequate availability of appropriate bedside instruments for evaluating respiratory mechanics in these patients [7].

Volumetric capnography (V_cap_) can provide clinically relevant volumetric parameters, such as the pulmonary elimination of carbon dioxide (CO_2_), respiratory dead space, and alveolar ventilation [8]. Several studies have demonstrated its efficacy for predicting acute respiratory distress syndrome-related mortality, diagnosis and therapeutic efficacy of pulmonary embolism, quantification of ventilation-perfusion mismatch, and PEEP titration [8]. Blankman et al. [9] reported that high PEEP levels increased the airway dead space volume (V_d,aw_) in a V_cap_ study of 15 adults. Furthermore, numerous animal studies on V_cap_ have indicated that high PEEP levels contribute to an increase in V_d,aw_ [10,11]. While some studies have investigated the clinical efficacy of V_cap_ in newborns, the limited clinical applicability for newborns, coupled with the requirement for specialized equipment with a significant apparatus dead space volume (V_d,app_), has resulted in minimal V_cap_ use among this population [12]. Although the molar mass signal from the ultrasonic flowmeter without an additional V_d,app_ is well suited for measuring dead space in infants, conventional ultrasonic flow sensors are too heavy and bulky for extensive clinical use in the NICU [12,13]. To our knowledge, no prior reports utilizing V_cap_ have described the impact of PEEP in infants.

In a prior study, we discussed a V_cap_ technique appropriate for newborns using a combination of ventilator and capnometer graphic waveforms [14,15,16]. However, this approach was not suitable for clinical application because it required the manual acquisition of partial pressure expiratory CO_2_ (P_E_CO_2_) and expired tidal volume (V_T,E_) data from the capnometer and respiratory ventilator waveforms. Therefore, we developed a novel V_cap_ technique for newborns that involved the automatic collection of P_E_CO_2_ and V_T,E_ data by linking a computer to a bedside monitor that aggregates all relevant information. This study aimed to assess the validity of this novel V_cap_ based on the patient monitor (V_cap,PM_) and explore the impact of PEEP on term and preterm infants.

## 2. Materials and Methods

### 2.1. Study Design

This single-center, prospective, nonrandomized, consecutive enrolment study was approved by the Kyoto Prefectural University of Medicine Clinical Ethics Committee in Kyoto, Japan (approval date 4 October 2021; approval number ERB-C-2145-1). This research was conducted in accordance with the Declaration of Helsinki. All methods were performed according to the relevant guidelines and regulations. Written informed consent was obtained from each participant’s legal guardians. We included patients receiving mechanical ventilation with a VN500 ventilator (Dräger Medical, Lübeck, Germany; maximum tidal volume [V_T_], 300 mL) using the synchronous intermittent mandatory ventilation mode. All patients had a stable general condition, especially in terms of respiratory status, and were deemed appropriate for inclusion by the attending clinicians. The exclusion criteria were as follows: >10% leaks, fighting the ventilator, pneumothorax, bronchial asthma, and respiratory tract disease. The respiratory circuits used in this study were selected according to body weight.

### 2.2. Methodology of V_cap,PM_

A schematic of the V_cap,PM_ is shown in Figure 1. A cap-ONE (TG-980P, Nihon Kohden, Tokyo, Japan; sampling frequency, 40 Hz; dead space, 1.8 mL) was placed between the flow sensor of the ventilator circuit and endotracheal tube. This lightweight mainstream capnometer is commonly used in NICUs and pediatric ICUs (PICUs). The cap-ONE and VN500 were both connected to the patient monitor (BSM-6000; Nihon Kohden, Tokyo, Japan); P_E_CO_2_ and respiratory flow data were obtained, which were upsampled to 125 Hz, in the patient monitor. We collected these data as CSV files and converted them into Excel files (Microsoft Excel, version 2409, Microsoft Corporation, Redmond, WA, USA) using a personal computer linked to the patient monitor. We then performed V_cap,PM_, for which P_E_CO_2_ was plotted against V_T,E_, as obtained by numerically integrating the flow from the beginning to the end of expiration. The Fowler dead space volume (V_d,Fowler_), the sum of V_d,aw_ and V_d,app_, was calculated from the resulting curve, as described previously [14,17]. V_d,app_ was calculated using the water displacement technique, while V_d,aw_ and alveolar V_T_ (V_A_) were computed as follows:V_d,aw_ = V_d,Fowler_ − V_d,app_(1)V_A_ = V_T_ − V_d,Fowler_(2)

The capnographic slopes of phases II (S_II_) and III (S_III_) were calculated by fitting linear regression lines over the volume-based capnograms and then normalized (Sn_II_ and Sn_III_) with V_T_ to account for anthropometric differences. We calculated the capnographic index (KPIv) as S_III_/S_II_ to quantify the degree of small airway obstruction [18]. Median values were calculated from ≥100 consecutive breaths and corrected for body weight at measurement.

### 2.3. Examination for Validity of V_cap,PM_ (Analysis 1)

To assess the accuracy of the V_cap,PM_ measurements, we compared the data obtained from both the V_cap,PM_ and HAMILTON-C1 (Hamilton Medical, Bonaduz, Switzerland), which was equipped with V_cap_ functionality. The CAPNOSTAT 5 (Philips, Philadelphia, PA, USA; sampling frequency, 100 Hz; dead space, 5 mL), a capnometer widely used in conjunction with the HAMILTON-C1, is not suitable for small infants primarily due to the impact of the V_d,app_. Therefore, Analysis 1 included ventilated newborns and children admitted to the NICU or PICU between December 2023 and August 2024 with body weights ranging from 3 to 12 kg, all of whom were suitable for the use of both the VN500 and HAMILTON-C1 (Figure 2). Upon the clinician’s assessment that extubation could be achieved, V_cap,PM_ was conducted, followed by V_cap_ with HAMILTON-C1 for 5 min. Subsequently, linear regression and Bland–Altman analyses were conducted to assess the accuracy of V_cap,PM_ for the median values of V_d,aw_, S_II_, S_III_, and P_E_CO_2_ derived from each method.

### 2.4. Impact of PEEP on Term and Preterm Newborns (Analysis 2)

Newborns admitted to the NICU between April 2022 and August 2024 were included; however, those with extremely low birth weight were excluded due to the possible adverse impacts of the cap-ONE V_d,app_ (Figure 3). Ventilator settings were adjusted to a fixed V_T_ of 5–7 mL/kg predicted body weight with a volume guarantee feature to circumvent a change in V_T_ due to differences in lung compliance at different PEEP levels. Throughout the procedure, the fraction of inspired oxygen (F_I_O_2_), inspiratory time, and respiratory rate remained consistent with baseline values established prior to starting. For cases in which the respiratory condition was stable, we maintained the PEEP at 5 cmH_2_O (mild level) for 15–20 min and incrementally increased it every 10 min to 7 cmH_2_O (moderate level) and 10 cmH_2_O or peak inspiratory pressure not exceeding 25 cmH_2_O (high level) during the execution of V_cap,PM_ (incremental phase). Following a 10 min stabilization period, the PEEP was reduced from high to moderate and mild levels (decremental phase), and the corresponding changes in the V_cap,PM_ measurements were assessed. Changes in V_cap,PM_ due to PEEP in term and preterm infants were compared during each phase.

### 2.5. Statistical Analyses

In Analysis 1, we evaluated the correlation between the V_cap,PM_ and HAMILTON-C1 measurements using Spearman’s correlation coefficient. The Bland–Altman technique was used to assess the agreement of the measurements by plotting the difference (D) of the average for each measurement. The 95% limits of agreement (D ± 2 standard deviations [SD]) depicted a range that included most of the differences in the methods. In Analysis 2, we first compared the V_d,aw_ and V_A_ at PEEP 5 cmH_2_O of term and preterm infants using the Mann–Whitney U test. Next, data on ventilator settings and V_cap,PM_ in each infant were compared among the three groups (mild, moderate, and high PEEP) using Friedman’s analysis of variance after Bonferroni adjustment for multiple comparisons. All the statistical analyses were performed using EZR software ver. 1.54 (Saitama Medical Centre, Jichi Medical University, Saitama, Japan).

## 3. Results

### 3.1. Analysis 1

The population in Analysis 1 consisted of 31 patients, including 8 newborns and 11 infants (mean age, 9 [interquartile range (IQR), 0–36] months; weight, 6.0 [IQR, 3.8–10.5] kg; sex, 39% male) (Table 1). The indications for intubation included asphyxia or encephalopathy (*n* = 17; 55%), respiratory failure (*n* = 7; 23%), and surgery (*n* = 7; 23%). The ventilator settings at the time of the investigation are shown in Table 1.

Figure 4 illustrates the significant correlations among V_cap,PM_ and HAMILTON-C1 (V_d,aw_: r = 0.99, *p* < 0.001; S_II_: r = 0.92, *p* < 0.001; S_III_: r = 0.90, *p* < 0.001; P_E_CO_2_: r = 0.92, *p* < 0.001). Bland–Altman plots revealed good agreement between the measurements, with a mean bias of 0.99 (V_d,aw_), 0.19 (S_II_), 0.01 (S_III_), and 1.1 (P_E_CO_2_); the 95% limits of agreement (± 2 SD) were 1.72 (V_d,aw_), 1.16 (S_II_), 0.09 (S_III_), and 2.27 (P_E_CO_2_).

### 3.2. Analysis 2

Analysis 2 included 28 term (median gestational age, 38 [IQR, 38–40] weeks; median birth weight, 2924 [IQR, 2725–3109] g) and 21 preterm (median gestational age, 33 [IQR, 31–34] weeks; weight, 1918 [IQR, 1356–2186] g) newborns, 41% of whom were male, with mean Apgar scores of 5 (IQR, 3–8) and 7 (IQR, 5–8) points at 1 and 5 min, respectively (Table 2). The median age at the time of measurement was 2 (IQR, 1–4) days in term and 4 (IQR, 3–5) days in preterm infants. The ventilator settings immediately before the analysis are presented in Table 2.

Table 3 shows the changes in different V_cap,PM_ parameters during the incremental phase. Despite there being no difference in V_T_, term infants exhibited lower V_d,aw_ and higher V_A_ at 5 cmH_2_O than preterm infants: V_d,aw_, 2.0 (IQR, 1.8–2.2) mL/kg vs. 2.6 (IQR, 2.2–2.8) mL/kg, respectively, *p* = 0.0037; V_A_, 3.6 (IQR, 3.2–4.2) vs. 3.1 (IQR, 2.7–3.7), respectively, *p* = 0.014. Additionally, although V_T_ remained constant throughout the procedure, high PEEP significantly increased V_d,aw_ and decreased V_A_ compared with mild PEEP in both term and preterm newborns. In the decremental phase, the restoration of a lower PEEP resulted in the normalization of both V_d,aw_ and V_A_ (Table 4). Furthermore, in both the incremental and decremental phases, the Sn_III_ and KPIv were greater in preterm newborns with mild PEEP compared to high PEEP. However, these differences were not evident in term patients. No complications, including changes in the percutaneous oxygen saturation, were detected during all procedures.

## 4. Discussion

In this study, we introduced a novel V_cap_ technique utilizing patient monitors and provided evidence of its consistency with the HAMILTON-C1 measurements. Furthermore, our findings indicate that higher PEEPs in newborns can be associated with increased airway dead space, resulting in a reduction in alveolar ventilation.

In this study, we found that the V_d,aw_ was 2.0 (IQR, 1.8–2.2) mL/kg in term newborns and 2.6 (IQR, 2.2–2.8) mL/kg in preterm newborns at a PEEP of 5 cmH_2_O. Although the normal range of V_d,aw_ in ventilated newborns remains controversial, premature infants reportedly have higher V_d,aw_. Dassios et al. [19] found a median V_d,aw_ of 2.4 (IQR, 1.9–2.9) mL/kg in term infants and 3.7 (IQR, 3.0–4.5) mL/kg in preterm infants. In another study, the V_d,aw_ was determined to be 1.47 ± 0.53 mL/kg for infants weighing ≥ 2500 g, compared with 1.84 ± 0.65 mL/kg for those with a weight ≤ 2500 g [20]. In preterm infants born between 22 and 32 weeks of gestation, during the end of the canalicular and saccular stages of lung development, the conducting airways are fully formed, and the formation of the respiratory bronchioles is nearly complete. However, these immature structures have limited support for collagen [21]. Additionally, alveolar proliferation and development occurred during the alveolar phase after 36 weeks [22,23]. Therefore, mechanical ventilation may induce airway dilation in premature infants with relatively stiff lungs.

To our knowledge, no previous study has demonstrated that high PEEP increases V_d,aw_ and reduces V_A_ in ventilated newborns, as reported in the present study, suggesting that airway dilation occurs during alveolar overdistension in these patients. While elevated levels of PEEP are generally associated with enhanced oxygenation, the findings of this study have indicated that high PEEP may also lead to hypercapnia. It is typically recommended that newborns be ventilated with a relatively low V_T_, between 4 and 5 mL/kg, to prevent lung injury [2]. Therefore, we suggest a higher respiratory rate setting be used when managing newborns with relatively high PEEP because of the reduction in effective alveolar ventilation. Furthermore, in this study, KPIv, which is widely used to evaluate ventilation inhomogeneity and ventilation-perfusion mismatch, increased at a PEEP of 5 cmH_2_O compared with higher PEEP in preterm infants [18,24]. Therefore, we believe that V_cap_, which can predict alveolar collapse and hyperinflation, enables the establishment of more accurate PEEP settings in intubated newborns.

The optimal level of PEEP for the neonates is defined as the level that provides the lowest F_I_O_2_ and acceptable blood gas and hemodynamic stability [25]. However, a recent systematic review concluded that evidence to guide PEEP level selection for preterm infants is insufficient [7]. Numerous strategies exist for determining bedside PEEP settings in adult patients, including V_cap_, mechanical parameters such as transpulmonary pressure, and imaging techniques such as computed tomography (CT) or electrical impedance tomography (EIT) [26]. However, there are a limited number of studies on PEEP titration in neonatal and pediatric populations. Although transpulmonary pressure measurements offer theoretical benefits in PEEP management, there are technical limitations to consider in young infants and children [27]. Furthermore, studies focusing on PEEP titration in children are scarce, and the only studies that have used transpulmonary pressure targeted PEEP titration are case reports [28,29]. EIT, a non-invasive imaging method for producing cross-sectional images of electrical conductivity distributions inside the human body, has been employed to titrate PEEP in children [30]. Despite its potential, it has not yet been widely adopted, particularly in NICUs. Recently, the utility of dynamic CT and flexible bronchoscopy techniques for PEEP titration in newborns has been demonstrated; however, the effects of PEEP on respiratory mechanics have not been demonstrated [31,32]. Dellacà et al. [33] demonstrated the feasibility of forced oscillation for non-invasive bedside PEEP titration in ventilated preterm newborns. Although this method can identify the optimal PEEP level, it requires equipment that is not widely used in the clinical setting, such as a servo-controlled linear motor. V_cap,PM_ has the potential to enhance suitable PEEP settings for newborns determination and contribute to lung protection strategies at a low cost because our methodology does not necessitate the use of specialized equipment.

This study has a few limitations. First, V_cap,PM_ was only validated for the combination of VN500 and cap-ONE. By assessing alternative combinations of ventilators and capnometers, our novel technique has the potential to yield a V_cap_ applicable to pediatric and adult patients undergoing ventilatory management in ICUs or operating rooms. Second, alveolar dead space (V_d,alv_), the volume of ventilated alveoli that did not receive blood flow and subsequently no gas exchange, was not measured in this study. A previous study on PEEP titration using V_cap_ showed that high PEEP increases the V_d,alv_. Consequently, more accurate PEEP titration in newborns may be accomplished by assessing the physiological dead space, the sum of V_d,Folwer_ and V_d,alv_. Finally, our study has statistical limitations. We excluded infants with unstable conditions, which may have introduced a potential selection bias. Additionally, the present study had a relatively small sample size. However, we performed a post hoc power analysis with a difference in the V_d,aw_ due to PEEP and found a power of 96%, which is significant.

## 5. Conclusions

We have shown that high PEEP induces airway dilation in newborns using a novel V_cap_ technique involving the automatic collection of P_E_CO_2_ and V_T,E_ data by linking a computer to the patient’s bedside monitor. We believe that this technique, without the need for special equipment, has the potential for wider clinical application in neonatal care.

## Figures and Tables

**Figure 1 biomedicines-13-02275-f001:**
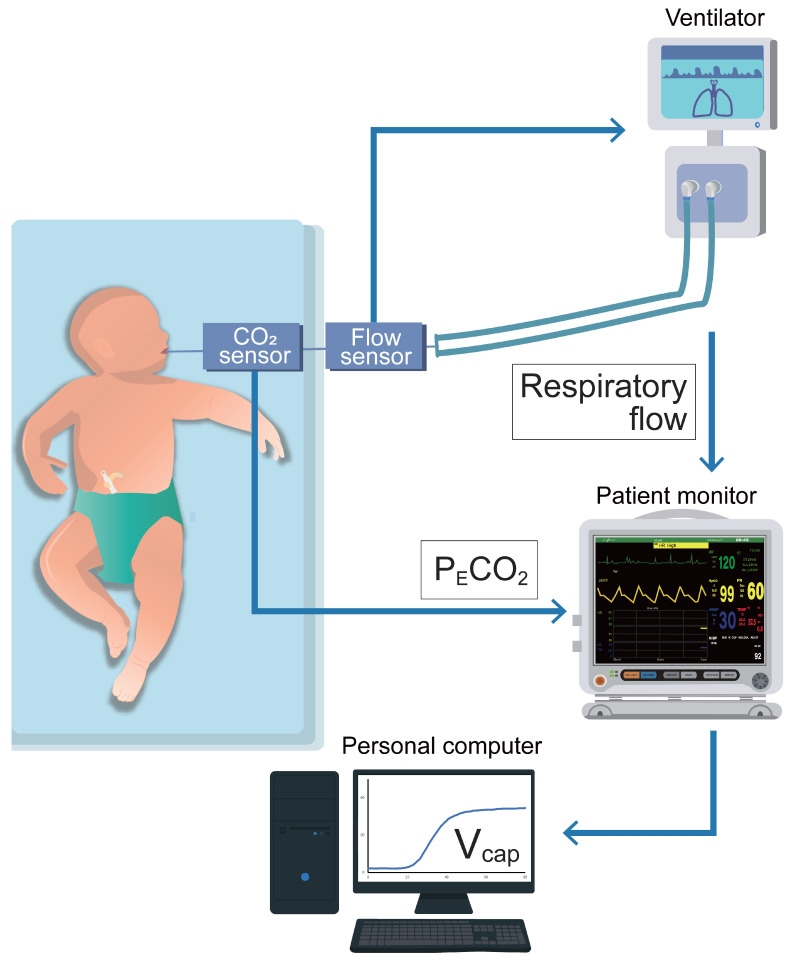
Schematic presentation of volumetric capnography based on patient monitor information. A capnometer and a ventilator were both connected to a patient monitor. Partial pressure expiratory carbon dioxide (P_E_CO_2_) and respiratory flow data were obtained using a personal computer linked to the patient monitor. Subsequently, we performed volumetric capnography based on patient monitor information, for which P_E_CO_2_ was plotted against expiratory tidal volume as obtained by numerically integrating the flow.

**Figure 2 biomedicines-13-02275-f002:**
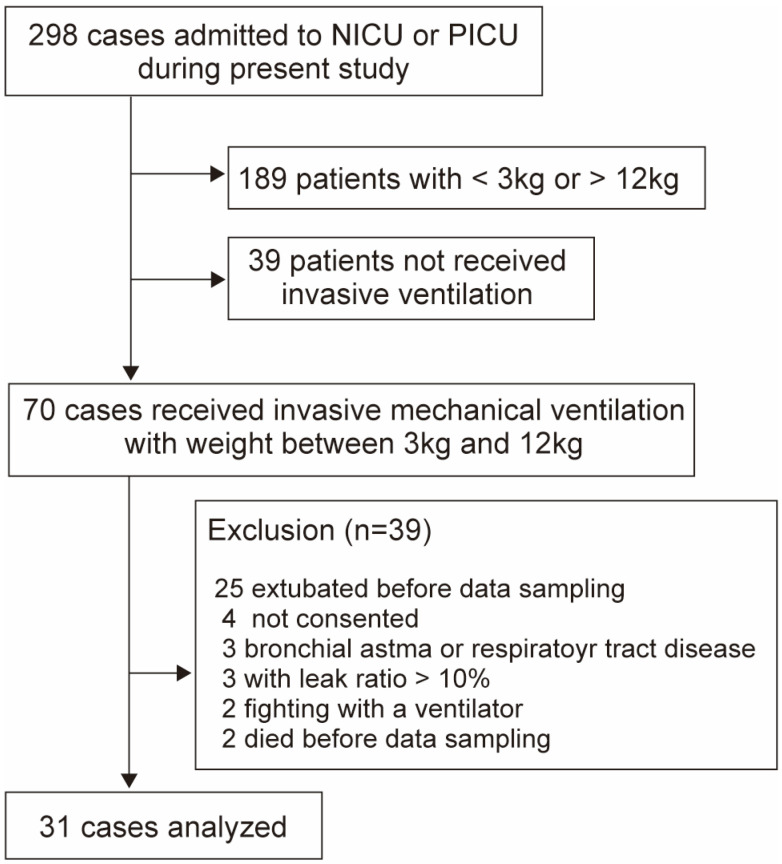
Flow diagram showing the number of included cases in analysis 1.

**Figure 3 biomedicines-13-02275-f003:**
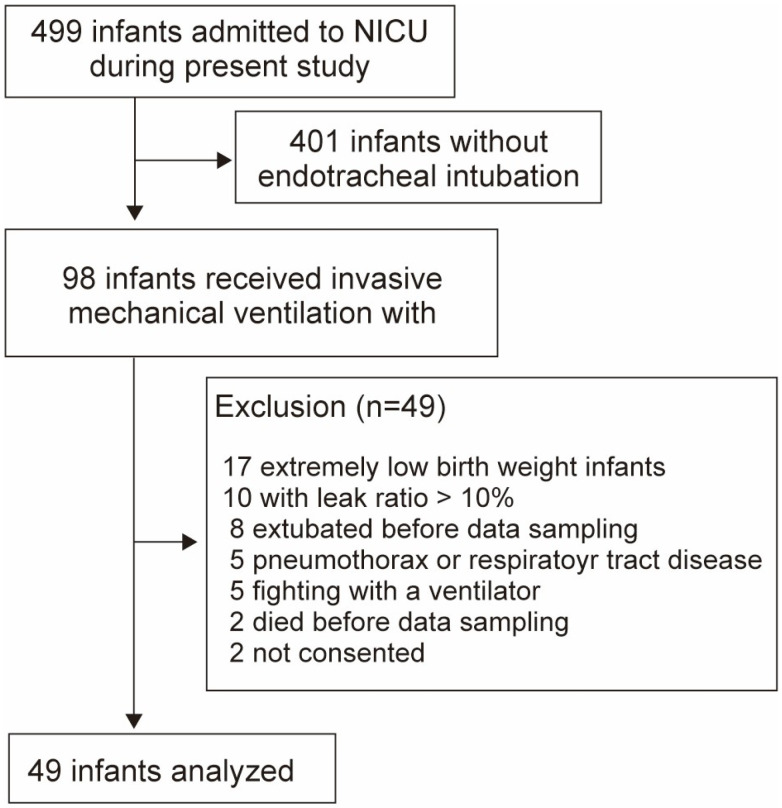
Flow diagram showing the number of included infants in analysis 2.

**Figure 4 biomedicines-13-02275-f004:**
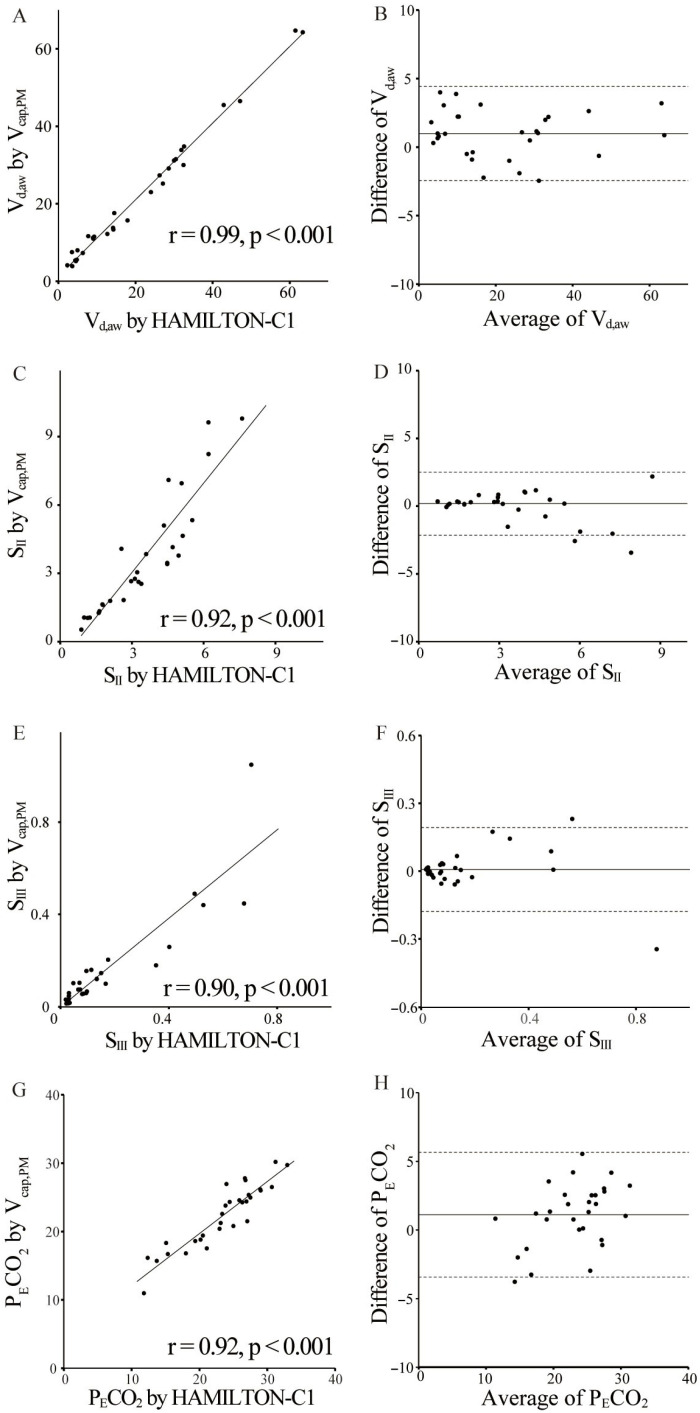
Validity of volumetric capnography based on patient monitor information. Scatter (**A**,**C**,**E**,**G**) and Bland–Altman (**B**,**D**,**F**,**H**) plots show agreement between measurements of volumetric capnography based on patient monitor information (V_cap,PM_) and HAMILTON-C1 for the airway dead space (V_d,aw_) (**A**,**B**), S_II_ (**C**,**D**), S_III_ (**E**,**F**), and partial pressure expiratory carbon dioxide (P_E_CO_2_) (**G**,**H**). There was a significant correlation between two methods (V_d,aw_: r = 0.99, *p* < 0.001; S_II_: r = 0.91, *p* < 0.001; S_III_: r = 0.90, *p* < 0.001; P_E_CO_2_: r = 0.92, *p* < 0.001). In the Bland–Altman analysis, the horizontal lines denote the estimated bias (solid) and the 95% limits of agreement (dotted line). The 95% limits of agreement were as follows: V_d,aw_, 0.99 ± 1.72; S_II_, 0.19 ± 1.16; S_III_, 0.01 ± 0.09; and P_E_CO_2_, 1.1 ± 2.7.

**Table 1 biomedicines-13-02275-t001:** Descriptive characteristics of the enrolled patients of analysis 1.

Parameter	Clinical Data of the Study Population (*n* = 31)
Age, months	9 (0–36)
Neonates, *n* (%)	8 (26)
Infants, *n* (%)	11 (35)
Children, *n* (%)	12 (39)
Weight, kg	6.0 (3.8–10.5)
Male/Female, *n*	12/19
Reason for intubation	
Asphyxia or encephalopathy, *n* (%)	17 (55)
Respiratory failure, *n* (%)	7 (23)
Operation, *n* (%)	7 (23)
Ventilator settings	
F_I_O_2_	0.21 (0.21–0.35)
PIP, cmH_2_O	17.6 ± 2.7
PEEP, cmH_2_O	5.9 ± 1.5
RR, /min	28 ± 7
MAP, cmH_2_O	9.0 ± 2.3
V_T_, mL/kg	9.8 ± 2.5

Values are represented as mean (±standard deviation) or median (interquartile range) unless specified otherwise. Abbreviations: F_I_O_2_, fraction of inspired oxygen; MAP, mean airway pressure; PEEP, positive end-expiratory pressure; PIP, peak inspiratory pressure; RR, respiratory rate; V_T_, tidal volume.

**Table 2 biomedicines-13-02275-t002:** Descriptive characteristics of the enrolled patients of analysis 2.

Parameter	Term Infants (*n* = 28)	Preterm Infants (*n* = 21)
Gestational age, weeks	38 (38–40)	33 (31–34)
Birth weight, g	2924 (2725–3109)	1918 (1356–2186)
Male/Female, *n*	12/16	8/13
Cesarean section, *n* (%)	20 (71)	18 (86)
Twin birth, *n* (%)	4 (14)	4 (19)
Apgar score at 1 min	5 (1–8)	5 (4–6)
Apgar score at 5 min	6 (4–9)	7 (6–8)
Postnatal surfactant, *n* (%)	4 (14)	10 (48)
Days of measurements, days	2 (1–4)	4 (3–5)
Ventilator settings at baseline		
F_I_O_2_	0.22 ± 0.01	0.23 ± 0.02
PIP, cmH_2_O	11.8 ± 1.4	12.9 ± 1.3
PEEP, cmH_2_O	5.4 ± 0.6	5.4 ± 0.7
RR, /min	38 ± 4	41 ± 6
MAP, cmH_2_O	6.7 ± 0.8	7.2 ± 0.9

Values are represented as means (±standard deviations) or median (interquartile range) unless specified otherwise. Abbreviations: F_I_O_2_, fraction of inspired oxygen; MAP, mean airway pressure; PEEP, positive end-expiratory pressure; PIP, peak inspiratory pressure; RR, respiratory rate.

**Table 3 biomedicines-13-02275-t003:** The impact of PEEP during incremental phase.

Parameter	Term (*n* = 28)			Preterm (*n* = 21)		
	Mild	Moderate	High	Mild	Moderate	High
PEEP, cmH_2_O	5.0 ± 0 ^†††^	7.0 ± 0 ^†††^	9.8 ± 0.4	5.0 ± 0 ^†††^	7.0 ± 0 ^†††^	9.6 ± 0.5
PIP, cmH_2_O	11.4 ± 1.0 ^†††^	15.3 ± 1.2 ^†††^	22.1 ± 1.8	12.4 ± 0.9 ^†††^	15.4 ± 0.8 ^†††^	22.5 ± 1.8
MAP, cmH_2_O	6.4 ± 0.3 ^†††^	8.8 ± 0.4 ^†††^	12.6 ± 0.5	6.8 ± 0.3 ^†††^	9.0 ± 0.4 ^†††^	13.0 ± 0.5
F_I_O_2_	0.22 ± 0.02	0.22 ± 0.02	0.22 ± 0.02	0.25 ± 0.01	0.25 ± 0.01	0.25 ± 0.01
SpO_2_, %	98.8 ± 1.4	98.7 ± 1.5	98.5 ± 1.4	96.7 ± 1.8	96.8 ± 1.8	96.9 ± 1.7
V_T_/kg, mL/kg	6.5 (5.5–6.6)	6.2 (5.4–6.6)	6.5 (5.5–6.7)	6.1 (5.6–6.8)	6.3 (5.6–6.5)	6.2 (5.4–6.5)
V_d,aw_, mL/kg	2.0 (1.8–2.2) ^†††^	2.1 (1.8–2.4) ^†††^	2.4 (2.2–2.7)	2.6 (2.2–2.8) ^†††^	2.8 (2.4–3.1) ^†††^	3.1 (2.5–3.4)
V_d,aw_/V_T_	0.34 (0.27–0.39) ^†††^	0.36 (0.30–0.42) ^†††^	0.40 (0.34–0.46)	0.40 (0.36–0.44) ^†††^	0.42 (0.39–0.45) ^†††^	0.48 (0.44–0.50)
V_A_, mL/kg	3.6 (3.2–4.2) ^††^	3.4 (3.1–3.9)	3.0 (2.7–3.8)	3.1 (2.7–3.7) ^†††^	2.9 (2.6–3.6) ^†††^	2.6 (2.2–3.0)
V_A_/V_T_	0.59 (0.53–0.64) ^†††^	0.56 (0.50–0.62) ^††^	0.52 (0.46–0.58)	0.50 (0.46–0.55) ^††††^	0.49 (0.45–0.52) ^†††^	0.43 (0.40–0.48)
S_II_, mmHg/mL	8.4 (6.4–9.5)	9.3 (6.1–10.5)	9.2 (6.6–10.6)	12.5 (9.8–15.1)	13.1 (10.3–18.2)	13.0 (10.5–18.3)
Sn_II_, mmHg	150 (123–173)	158 (124–174)	166 (139–194)	135 (117–161)	146 (138–174)	156 (138–174)
S_III_, mmHg/mL	0.17 (0.12–0.26)	0.18 (0.08–0.26)	0.2 (0.11–0.33)	0.37 (0.18–0.68)	0.35 (0.20–0.71)	0.22 (0.09–0.44)
Sn_III_, mmHg	3.9 (1.9–4.5)	3.2 (1.6–4.1)	3.6 (2.2–5.0)	4.7 (2.7–8.8) ^†^	4.2 (2.7–9.8)	2.3 (1.5–3.7)
KPl_V_	24 (12–35)	20 (11–30)	22 (11–33)	35 (17–54) ^†^	28 (17–58)	16 (9–26)

Values are represented as mean ± standard deviation or median (interquartile range) unless specified otherwise. Abbreviations: F_I_O_2_, fraction of inspired oxygen; KPIv, capnographic index; MAP, mean airway pressure; PEEP, positive end-expiratory pressure; PIP, peak inspiratory pressure; Sn_II_, normalized S_II_; Sn_III_, normalized S_III_; SpO_2_, percutaneous oxygen saturation: V_A_, alveolar tidal volume; V_d,aw_, airway dead-space; V_T_, tidal volume. The difference among the groups was tested for significance with Friedman’s analysis of variance after Bonferroni adjusted. ^†^ <0.05 versus high; ^††^ <0.01 versus high; ^†††^ <0.001 versus high; ^††††^ <0.0001 versus high.

**Table 4 biomedicines-13-02275-t004:** The impact of PEEP during decremental phase.

Parameter	Term (*n* = 28)			Preterm (*n* = 21)		
	High	Moderate	Mild	High	Moderate	Mild
PEEP, cmH_2_O	9.8 ± 0.4	7.0 ± 0 ^†††^	5.0 ± 0 ^†††^	9.6 ± 0.5	7.0 ± 0 ^†††^	5.0 ± 0 ^†††^
PIP, cmH_2_O	22.0 ± 1.6	15.3 ± 1.0 ^†††^	11.4 ± 0.9 ^†††^	22.4 ± 1.7	15.2 ± 0.7 ^†††^	12.2 ± 0.8 ^†††^
MAP, cmH_2_O	12.7 ± 0.5	8.9 ± 0.4 ^†††^	6.4 ± 0.3 ^†††^	13.0 ± 0.5	9.0 ± 0.3 ^†††^	6.7 ± 0.3 ^†††^
F_I_O_2_	0.22 ± 0.02	0.22 ± 0.02	0.22 ± 0.02	0.23 ± 0.01	0.23 ± 0.01	0.23 ± 0.01
SpO_2_, %	99.0 ± 1.3	98.7 ± 1.6	98.7 ± 1.6	98.1 ± 2.0	97.7 ± 2.0	97.1 ± 2.0
V_T_/kg, mL/kg	6.5 (5.5–6.6)	6.5 (5.5–6.8)	6.5 (5.6–6.9)	6.2 (5.7–6.9)	6.9 (6.3–7.5)	6.1 (5.5–7.4)
V_d,aw_, mL/kg	2.4 (2.2–2.4)	2.1 (1.9–2.3) ^†††^	2.0 (1.8–2.2) ^†††^	3.2 (2.5–3.5)	2.6 (2.3–3.0) ^†††^	2.5 (2.1–2.7) ^†††^
V_d,aw_/V_T_	0.40 (0.34–0.46)	0.36 (0.31–0.39) ^†††^	0.33 (0.29–0.37) ^†††^	0.48 (0.44–0.5)	0.39 (0.33–0.44) ^†††^	0.39 (0.34–0.42) ^†††^
V_A_, mL/kg	2.9 (2.6–3.7)	3.1 (2.9–3.9)	3.4 (2.9–4.1) ^†^	2.6 (2.2–3.0)	3.5 (2.7–3.7) ^†††^	3.3 (2.8–4.0) ^†††^
V_A_/V_T_	0.52 (0.46–0.58)	0.55 (0.52–0.60) ^††^	0.56 (0.51–0.62) ^†††^	0.42 (0.40–0.47)	0.52 (0.47–0.57) ^†††^	0.52 (0.48–0.57) ^†††^
S_II_, mmHg/mL	9.2 (6.6–10.8)	8.6 (6.1–10.3)	8.5 (7.0–9.2)	13.0 (10.4–18.3)	13.5 (9.5–15.7)	12.8 (9.3–15.4)
Sn_II_, mmHg	166 (139–193)	155 (124–176)	152 (125–164)	156 (138–174)	162 (130–191)	137 (125–164)
S_III_, mmHg/mL	0.2 (0.11–0.32)	0.15 (0.09–0.19)	0.17 (0.06–0.29)	0.22 (0.09–0.44)	0.28 (0.12–0.62)	0.42 (0.25–0.77)
Sn_III_, mmHg	3.6 (2.2–4.9)	3.0 (1.7–4.0)	3.5 (1.1–5.2)	2.3 (1.5–3.7)	4.3 (1.7–7.1)	4.7 (3.1–10.1) ^†^
KPl_V_	22 (11–34)	20 (11–23)	24 (7–34)	16 (9–27)	20 (10–59)	33 (23–58) ^†^

Values are represented as mean ± standard deviation or median (interquartile range) unless specified otherwise. Abbreviations: F_I_O_2_, fraction of inspired oxygen; KPIv, capnographic index; MAP, mean airway pressure; PEEP, positive end-expiratory pressure; PIP, peak inspiratory pressure; Sn_II_, normalized S_II_; Sn_III_, normalized S_III_; SpO_2_, percutaneous oxygen saturation: V_A_, alveolar tidal volume; V_d,aw_, airway dead-space; V_T_, tidal volume. The difference among the groups was tested for significance with Friedman’s analysis of variance after Bonferroni adjusted. ^†^ <0.05 versus high; ^††^ <0.01 versus high; ^†††^ <0.001 versus high.

## Data Availability

The datasets used and/or analyzed during the current study are available from the corresponding author on reasonable request.

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
