# Peer review of "High PEEP Increases Airway Dead Space and Decreases Alveolar Ventilation: A New Technique for Volumetric Capnography"

_biomedicines, 2025, doi:10.3390/biomedicines13092275_

Round 1
Reviewer 1 Report
Comments and Suggestions for Authors
This study investigates the impact of PEEP on airway dead space and alveolar ventilation in neonates using a novel technique for volumetric capnography (Vcap). This is a well-designed and clinically important study with high potential impact.
Here are some suggestions for the authors:
- Novelty: The study introduces an innovative, practical method (Vcap).
- Methodology: It is a well-designed study using existing technology.
- Results: The results are well-presented.
- Discussion: The discussion could compare findings with recent neonatal ventilation guidelines and prior Cochrane reviews to contextualize clinical implications.
- Limitations: The authors have excluded unstable infants, and this could be considered a potential selection bias.
Recommendations
- Expand the discussion to better situate findings within neonatal ventilation literature and emphasize clinical translation.
- Address limitations regarding selection bias and operator variability.
Author Response
Thank you very much for your helpful comments. In accordance with Reviewer’s comments, we added following sentences with a new reference.
"The optimal level of PEEP for the neonates is defined as the level that provides the lowest FIO2 and acceptable blood gas and hemodynamic stability [25]. However, a recent systematic review concluded that evidence to guide PEEP level selection for preterm infants is insufficient [7]. "(page 9, line 261-264)
the new reference:
[25] Sweet, D.G.; Carnielli, V.; Greisen, G.; Hallman, M.; Ozek, E.; Te Pas, A.; Plavka, R.; Roehr, C.C.; Saugstad, O.D.; Simeoni, U.; Speer, C.P.; Vento, M.; Visser, G.H.A.; Halliday, H.L. European consensus guidelines on the management of respiratory distress syndrome-2019 update. Neonatology 2019, 115, 432-450. https://doi.org/10.1159/000499361
In addition, we added following sentences to the limitation section.
"Finally, our study has statistical limitations. We excluded infants with unstable conditions, which may have introduced a potential selection bias." (page 9, line 292-294)
Reviewer 2 Report
Comments and Suggestions for Authors
An interesting study measuring the change in dead space and airway diameter with changes in of PEEP from 5 to 10 cm in ventilated neonates and infants.
The authors developed their own unique methodology to accurately measure Exhaled CO2
They have demonstrated findings consistent with previous studies that the increases in PEEP within the clinical ranges used for ventilated neonates does increase dead space and decrease alveolar ventilation
The changes observed are statistically significant but it remains to be seen if there are clinical consequences when using the higher PEEP ranges studies in this manuscript, that outweighs the improvements in oxygenation - they note that their results suggest the use of an increased ventilator rate when using higher PEEP ranges to offset the fall in alveolar ventilation.
The use of higher PEEP levels does occur and this study suggests caution with that approach and a careful assessment of ventilation and oxygenation with higher PEEP levels
Author Response
Thank you very much for your helpful comments. In accordance Reviewer's comments, we added following sentence to the Discussion section.
"While elevated levels of PEEP are generally associated with enhanced oxygenation, the findings of this study have indicated that high PEEP may also lead to hypercapnia." (page9, line 250-252)